# Evaluation and Comparison of Genomic DNA Extraction Methods and PCR Optimization on Archival Formalin-Fixed and Paraffin-Embedded Tissues of Oral Squamous Cell Carcinoma

**DOI:** 10.3390/diagnostics12051219

**Published:** 2022-05-12

**Authors:** Samar Saeed Khan, Manisha Tijare, Sowmya Kasetty, Megha Jain, Ahmed Alamoudi, Hammam Ahmed Bahammam, Sarah Ahmed Bahammam, Maha A. Bahammam, Saranya Varadarajan, A. Thirumal Raj, Shankargouda Patil

**Affiliations:** 1Division of Maxillofacial Surgery and Diagnostic Sciences, Department of Oral and Maxillofacial Pathology, College of Dentistry, Jazan University, Jazan 45142, Saudi Arabia; samarkhan8@gmail.com; 2Department of Dentistry, Government Medical College and Hospital, Gondia 441601, India; manisha.tijare@gmail.com; 3Oral Pathology Division, Oral Basic and Clinical Sciences, College of Dentistry, Qassim Private College, Buraidah 52571, Saudi Arabia; sowmyakasetty@gmail.com; 4Department of Dentistry, Chhindwara Institute of Medical Sciences, Chhindwara 480001, India; megha.vipin12@gmail.com; 5Department of Oral Biology, College of Dentistry, King Abdulaziz University, P.O. Box 80209, Jeddah 21589, Saudi Arabia; ahmalamoudi@kau.edu.sa; 6Department of Pediatric Dentistry, College of Dentistry, King Abdulaziz University, P.O. Box 80209, Jeddah 21589, Saudi Arabia; habahammam@kau.edu.sa; 7Department of Pediatric Dentistry and Orthodontics, College of Dentistry, Taibah University, Universities Road, P.O. Box 344, Medina 46526, Saudi Arabia; sbahammam@taibahu.edu.sa; 8Department of Periodontology, Faculty of Dentistry, King Abdulaziz University, P.O. Box 80209, Jeddah 21589, Saudi Arabia; mbahammam@kau.edu.sa; 9Executive Presidency of Academic Affairs, Saudi Commission for Health Specialties, Riyadh 11614, Saudi Arabia; 10Department of Oral Pathology and Microbiology, Sri Venkateswara Dental College and Hospital, Chennai 600130, India; vsaranya87@gmail.com (S.V.); thirumalraj666@gmail.com (A.T.R.); 11Department of Maxillofacial Surgery and Diagnostic Science, Division of Oral Pathology, College of Dentistry, Jazan University, Jazan 45142, Saudi Arabia

**Keywords:** oral squamous cell carcinoma, formalin-fixed tissues, formalin-fixed paraffin-embedded tissues, deparaffinization, polymerase chain reaction

## Abstract

Recovery and amplification of nucleic acids from archived formalin-fixed tissue samples is the most developing field in retrospective genetic studies. We compared different deparaffinization methods and DNA isolation techniques, and intergroup comparisons were performed to evaluate the effectiveness of different storing methods for archival OSCC samples based on obtained mean DNA quantity, quality, and PCR amplification of the P53 gene. The study comprised 75 archival histologically diagnosed OSCC samples which were divided into Group I: Formalin-fixed paraffin-embedded tissue blocks and Group II: Long-term formalin-fixed tissue. A comparison of different deparaffinization methods showed that xylene deparaffinization is an efficient method to obtain suitable DNA. Comparing different DNA isolation techniques illustrated that the conventional phenol–chloroform method gives better integrity to DNA in contrast with the kit method. Comparison between FFPET and long-term FFT samples demonstrated that samples fixed in formalin overnight and embedded in wax yield better quality and quantity DNA in comparison with long-term samples fixed in formalin. To obtain suitable integrity of DNA, tissue samples should be stored by fixing in formalin overnight followed by preparation of paraffin tissue blocks, deparaffinization by xylene, and subjecting them to the conventional phenol–chloroform DNA isolation protocol.

## 1. Introduction

Archives of formalin-fixed tissues (FFT) and formalin-fixed paraffin-embedded tissue (FFPET) represent a remarkable source for morphologically well-defined tissues, serving as a valuable source of retrospective biological material for research and to the corresponding molecular findings with treatment and clinical follow-up of the disease process [1,2]. The majority of archival samples are stored in formalin, which enhances better tissue handling properties. It improves the chance of storing for the long term with optimal histological quality and makes them available in substantial quantities for minimal price [1,3,4].

In recent times, the disease process is being better understood through the application of molecular techniques, as changes at the molecular level precede clinical alteration [5]. The molecular approach in genetics will serve in determining chromosomal alterations and identifying genes disrupted in a variety of diseases, including cancer. Particularly in oral cancer, molecular studies serve to elevate clinical assessment, classify oral lesions, and predict the potential of malignant transformation of oral lesions, thereby increasing the ambit to make an early diagnosis and begin oral cancer treatment [6,7].

Despite many problems associated with FFT and FFPET, formalin-fixed specimens are kept in collection around the globe, and the genetic information of humans and pathogens are often critical in medical investigations. Thus, methods are being discovered in obtaining optimal quality of nucleic acids for retrospective studies. The successful extraction of quality DNA from fixed specimens depends on many parameters that cause the nucleic acids to degrade, including pre-fixation factors such as the type and amount of tissues and autolysis, fixation-related factors such as temperature, pH, concentration, and time, and post-fixation factors such as time of storage and processing [8,9]. Irrespective of the DNA technique used for archival FFPET samples, complete deparaffinization is mandatory as improper removal of paraffin wax will hinder the DNA isolation procedure. Therefore, an effective and reliable protocol for deparaffinization should be set as an important process to obtain suitable DNA integrity. 

The recent trends in investigation procedures are using polymerase chain reaction (PCR) to analyze the changes at a molecular level in progression and detection of several diseases including cancer [10]. The high molecular weight DNA is usually not necessary for successful amplification in PCR and therefore this method is ideally suited for DNA templates obtained from formalin-fixed paraffin wax-embedded archival material [11].

Very few studies have evaluated the efficacy of sample storage on DNA yield. Therefore, proper criteria should be established regarding the preservation of samples, whether to preserve tissue in formalin for the long term or to process routinely and make paraffin tissue blocks. Many studies have been undertaken in the past for comparison of different deparaffinization and DNA extraction techniques in various carcinomas, such as cervical cancer, lymphoma, hepatocellular carcinoma, stomach adenocarcinoma, and colonic tumor [11,12,13]. To our knowledge, very few to no studies have been performed on oral squamous cell carcinoma (OSCC) studies with this background. 

The current study was designed to evaluate the efficiency of different deparaffinized and DNA extraction methods from FFPET to assess the quality and quantity of DNA from long-term formalin-fixed specimens with that of archival paraffin tissue blocks. Further, we assess the viability of the obtained genomic DNA using the PCR technique for p53 gene amplification in OSCC.

## 2. Materials and Methods

Archives of FFT and FFPET samples (2003–2011) histologically diagnosed as OSSC were retrieved and included in the study (Table 1). The study was approved by the Institutional Ethics Committee (PU/CSRD/DIR/4305) of People’s University, Centre of Scientific Research and Development, Bhopal.

### 2.1. Sample Collection

For Group I: Paraffin-embedded tissue (*n* = 15) was collected by sectioning it into 10 µm thickness using a semi-automatic microtome. To eliminate cross-contamination, microtome blades were cleaned with disinfectant and xylene every time. The sectioned areas were carefully examined to make sure an equal amount of the tissue in each set was kept constant. The collected sections were subjected to deparaffinization using xylene and the heating method. Simultaneously, the DNA extraction procedure was carried out using the conventional phenol–chloroform and DNA kit method. Later, a comparison was undertaken between different deparaffinization methods and the DNA yield from the different DNA isolation procedures.

For Group II: The long-term formalin-fixed tissues were recorded after freezing with liquid nitrogen using a mortar and pestle and pulverization. The obtained tissue powder was investigated for DNA yield using the phenol–chloroform and kit method. Finally, the difference in deparaffinization methods for the DNA yield by different DNA isolation procedures were assessed.

### 2.2. Deparaffinization Process

Deparaffinization of paraffin-embedded tissues: Sections were deparaffinized by two methods: (i) the xylene method and (ii) the heating method. In the xylene method, 1 mL of xylene was added to microcentrifuge tubes containing sectioned paraffin-embedded tissue. The tubes were vortexed for 5 min and kept at 60 °C in a water bath for 30 min. Then, samples were centrifuged for 3–4 min at 12,000 rpm. The above-mentioned steps were repeated 2–3 times till clear supernatant was obtained and further taken for DNA isolation. Meanwhile, in the heating method, paraffin sections were taken on simple non-albumin-coated slides and kept on the hot plate for deparaffinization and were heated till all sections were deparaffinized. Wax residue on slides was manually removed with blotting paper to ensure complete deparaffinization and the sections were transferred into fresh microcentrifuge tubes to isolate the DNA.

### 2.3. DNA Isolation by Phenol–Chloroform Method

A standard quantity of 25 mg of tissues was taken for the DNA isolation procedure. The obtained tissue pellet was subjected to a series of chilled graded ethanol (50%, 70%, and 100%) after deparaffinization for 10 min in a microcentrifuge machine at 12,000 rpm. The pulverized samples of formalin-fixed tissue were taken and directly subjected to the digestion method. In the digestion method, DNA extraction buffer of 500 µL (1 M NaCl pH 8, 0.5 EDTA pH 8, 1 M Tris HCL pH 8, 10% SDS) and proteinase K of 40 µL were mixed with the sample. Tubes were swirled shortly and then incubated overnight at 55–56 °C. When sufficient protein digestion was achieved, it was followed by inactivating proteinase K by heating for 15 min at 85 °C, and later deparaffinized and digested samples were taken for a further DNA isolation procedure. Following the inactivation of proteinase K, 0.5 mL saturated phenol pH 8 was added, and tubes were slowly shaken for 5 min and centrifuged at 8000 rpm for 5 min. The collected supernatant was mixed with 220 µL of phenol: chloroform:isoamyl alcohol (25:24:1) and centrifuged at 8000 rpm for 5 min to achieve an aqueous phase. Then, the supernatant was transferred into new centrifuge tubes and the above (phenol: chloroform: isoamyl alcohol) steps were repeated 2–3 times. In the obtained supernatant, 3 M sodium acetate and 100% ethanol (3 times the supernatant volume) were added and kept overnight to precipitate DNA at −20 °C. Precipitated DNA was collected for 20 min at 4 °C at 8000 rpm by centrifugation, 70% ethanol was used for washing, and it was air-dried at room temperature. The DNA pellet was re-suspended in a 20 µL TE buffer. 

### 2.4. DNA Isolation by Kit Method

A standard quantity of 25 mg of tissues was taken for the DNA isolation procedure. Samples were placed into a microcentrifuge tube. For deparaffinization, 1 mL of xylene was added to samples. They were rocked at room temperature for 5 min and centrifugation was performed for 3 min at 14,000 rpm. Then, the supernatant was discarded, and the above steps were repeated 3–4 times to ensure the removal of paraffin wax. Further DNA isolation (HIPURATM PET) and purification (spin kit Himedia-MB530) was performed following the manufacturer’s protocol using the specified quantity and concentration of given solutions provided in the kit. The obtained DNA was then stored at −20 °C for PCR analysis.

### 2.5. Quantitative and Qualitative Assessment of Extracted DNA

The quality and quantity of extracted DNA by both the methods was assessed using Picodrop UV-Spectrophotometer (PICOPET01). Exactly, 1 µL of DNA sample taken was placed in a cuvette of Picodrop UV-Spectrophotometer, and the absorbance was measured at 260 and 280 nm. Then, the ratio of absorbance was used to determine the purity of DNA samples. The presence of nucleic acid was indicated by a ratio obtaining between 1.6 and 1.8, whereas a ratio less than 1.6 would indicate protein contamination and more than 1.8 would indicate phenol and chloroform contamination and/or RNA content. The measurement was repeated by taking triplicates of each sample in a group.

### 2.6. PCR Analysis of Obtained DNA

PCR analysis using the p53 gene was conducted to assess the viability of obtained DNA and to evaluate the amplification property and whether acquired DNA was valuable for further molecular studies. A total of 25 μL PCR reaction mixture containing reaction components and primers (p53-Forward primer: 5′-CCTATCCTGAGTAGTGGTAATCTAC-3′, p53-Reverse primer: 5′-GTCCTGCTTGCTTACCTCGCTTAGT-3′) was prepared for PCR amplification and programmed in an Eppendorf master cycler gradient. The analysis was performed in accordance with the standardized protocol given as detailed in Table 2.

### 2.7. Gel Electrophoresis and Analysis of PCR Product

Agarose gel electrophoresis was carried out for qualitative analysis of samples prepared. The PCR products were loaded carefully into the well of the casted gel. Each PCR was conducted as an experiment, with controls (distilled water instead of template DNA) to test the purity and viability of reagents. The analysis was performed for all the samples at least three times with each selected primer to check the reproducibility. A DNA ladder was also loaded along with the samples to quantify DNA and electrophoresis was carried out at a constant voltage of 70 V till the dye had run three-quarters of the distance on the gel. Further analysis of the gel was conducted using the gel documentation system; after running the gel, it was placed on the gel documentation system and was visualized by 302 nm high intensity UV light. The image was captured and analyzed using Quantity One Software, and the molecular weight was calculated using this software. 

### 2.8. Analysis of PCR Data

Clear and separated amplified fragments from all primers were scored by visual observations for their presence/absence, respectively, in binary form by denoting ‘1’ and ‘0’ by following the molecular size using the gene ruler low range DNA Ruler Plus, which was run along with the amplified products. The data obtained were then used as input for further analysis.

### 2.9. Statistical Analysis

Statistical analysis was undertaken to establish correlation amongst study groups for qualitative and quantitative assessment of obtained DNA. Statistical differences were assessed by the Student *t*-test, Z test, and analysis of variance (one-way ANOVA) with Tukey’s honest significant difference post hoc analysis using SPSS software.

## 3. Results

The study was designed in three different phases. In the first phase, the study determined the potential of different deparaffinizing techniques from archival FFPET. The second phase showed an intra-group comparison of obtained DNA based on quantitative and qualitative analysis. The third phase of this study consisted of an inter-group that was the archival FFPET and FFT assessment of obtained DNA quantity and quality. Lastly, as a part of the study, the obtained DNA from archival OSSC samples was taken up for PCR reaction using the p53 gene to determine whether obtained DNA from different extraction protocols was suitable for successful amplification.

### 3.1. Assessment of Quantity and Quality of Obtained DNA in Each Study Group

The mean DNA quantity and quality obtained from the study groups are depicted in Table 3.

### 3.2. Assessment Based on Deparaffinizing Techniques

The potential of differences in deparaffinizing techniques were assessed from the archival FFPET. Two different techniques, such as the xylene and heating method, were adopted in the collected FFPET. Analysis of DNA quantity based on the deparaffinizing methods using a *t*-test showed significant difference between the methods, suggesting the xylene method is more suitable for a high yield of DNA. Alternatively, no significance was achieved on DNA purity between the methods (Table 4).

### 3.3. DNA Quantity Differences within the Group

Comparison of mean DNA quantity obtained in Group I samples (FFPET) were found to be statistically significant using one-way ANOVA based on post hoc analysis. This showed that mean DNA quantity in Sub-Group IA was significantly higher as compared to Sub-Group IB and IC. Though the mean obtained DNA quantity was higher in Sub-Group IB in comparison with Sub-Group IC, results between them were non-significant (Table 5). Likewise, on comparing mean DNA quantity obtained in Group II samples (FFT) using the Student *t*-test, it was found to be statistically significant. Results showed that mean DNA quantity in Sub-Group IIA was significantly higher as compared to Sub-Group IIB (Table 5). 

### 3.4. DNA Quantity Differences between the Groups

Comparison of mean DNA quantity obtained through conventional extraction (Group IA, IB, and IIA) between FFPET and FFT was found to be statistically significant using a one-way ANOVA test. To establish significant correlation amongst different study groups, Tukey’s HSD statistical analysis was performed. Results showed that mean DNA quantity in Sub-Group IA was significantly higher as compared to Sub-Group IB and IIA. Though mean obtained DNA quantity was higher in Sub-Group IB in comparison with Sub-Group IIA, results between them were non-significant (Table 6). Similarly, the mean DNA quantity obtained by HiPurATM Paraffin-Embedded Tissue DNA Purification Spin Kit in (Group IC and IIB) samples when compared was found to be statistically significant using a *t*-test. Results showed that mean DNA quantity in Sub-Group IC was significantly higher in comparison with Sub-Group IIB (Table 6).

### 3.5. Comparison of DNA Quality within the Groups

For quality of DNA obtained from FFPET (Group I) when subjected to the ‘Z’ test, there was a significant relationship observed between the DNA qualities of Sub-Group IA and IC. DNA quality was found to be significantly better in Sub-Group IA than IC. However, no significant difference was found between Sub-Group IA and IB and between Sub-Group IB and IC. Mean DNA quality in Group II (archival FFT) when compared was found to be better in Sub-Group IIA than IIB. When subjected to the ‘Z’ test, no significant relationship was seen between the DNA qualities of Sub-Group IIA and IIB (Table 7). 

### 3.6. Comparison of DNA Quality between the Groups

A comparison of mean DNA quality obtained through the conventional extraction method in Study Group IA and IB with Study Group IIA was performed. To establish a significant correlation among different study groups a “Z test” was applied. Results showed that DNA quality in Sub-Group IA was significantly higher in comparison with Sub-Group IIA. No significant relationship was established between Sub-Groups IA and IB and also between Sub-Group IB and IIA. Likewise, the comparison based on the DNA quality obtained through HiPurATM from the kit method was evaluated. The Group IC and IIB samples assessed using the Z-test were found to be statistically non-significant. Results showed that DNA quality in Sub-Group IC was better in comparison with Sub-Group IIB (Table 8).

### 3.7. Assessment and Validation Based on PCR Amplification

PCR analysis was undertaken to assess the viability of obtained DNA by evaluating the amplification ability through the well-reported p53 gene in cancer. Analysis showed PCR amplification of the p53 gene was found to be highest in Sub-Group IA followed by Sub-Group IB, IC, IIB, and IIA, respectively (Table 9).

## 4. Discussion

Almost all samples stored worldwide in hospital pathology are FFPET for sample preservation and archiving. It serves as a valuable source of retrospective biological material for future research and for correlating molecular findings with therapy and prognosis [1]. The use of paraffin-embedded tissues in PCR-based studies resulted in many exciting new insights in the areas of cancer research, genetic and infectious disease, and molecular epidemiology. A greater amount of FFPET has been used for diagnosing surgical pathology. The success of amplification of DNA from PET relies on various factors including fixation time, the type of the fixative used, designed primer of choice, storage time, and, importantly, conditions of PCR. Few studies have shown molecule yield differences based on the storage duration [10,14,15,16].

Additionally, the use of FFPE tissue samples has some limitations in molecular pathology, as formalin causes degradation and fragmentation of DNA resulting in low quantity and poor quality of obtained DNA through the following changes [9]: (i) During the fixation process, the formaldehyde would react with the amino group of guanine (G), cytosine (C), and adenine (A), forming the covalent linkages which in turn cross-link with proteins, leading to formation of RNA and DNA protein cross-linkages. (ii) The nucleic acid fragmentation may occur in FFT due to aging of the specimen or in conditions when unbuffered fixative solution is used, and where the pH is less than one. (iii) The extended fixation intervals would produce lesser PCR yields and lower the ability to amplify longer templates. (iv) Formalin residuals inhibit enzyme activity during the extraction and PCR reaction.

The DNA extracted from FFPET samples was discovered in 1985 [17]. The quality and quantity of extracted DNA, and the subsequent DNA amplification success, depends on various factors during, before, and after extraction. They are not only limited to the amount and type of the tissue, but also to the type of fixatives used to preserve the tissues, the fixation duration, paraffin block age, and conditions used for storage, along with the length of segment that has to be amplified from the desired DNA. 

This study aimed to compare the efficacy of the deparaffinizing technique, as paraffin removal from the tissues is one of the crucial steps for optimal DNA extraction since undissolved paraffin could produce a low quality of sample and inhibit PCR [12]. Stanta et al. found that eliminating the deparaffinization method leads to PCR inhibition and therefore is an important step, whereas according to Gilbert MT et al., Shri SR et al., and Lin W et al., removal of paraffin is unnecessary and its omission takes place during tissue processing without affecting the results [9,18,19,20]. Results from the current study show the samples of deparaffinization in xylene gave significantly increased quantity and better quality of DNA in comparison with samples deparaffinized by heating; thus, suggesting that xylene is a better deparaffinizing agent against heating. 

Xylene is a commonly used clearing agent miscible with most organic solvents and paraffin and therefore commonly used as a deparaffinizing agent in laboratories. The use of xylene as a deparaffinizing agent for DNA isolation was firstly described by Goelz et al. and since then many studies were conducted using xylene and found it to be suitable for a DNA isolation protocol [14,17,21,22,23,24]. In the present study, deparaffinization by xylene with two subsequent washes and incubating it at 60 °C for 60 min each, has given the best yield of DNA, which is in favor with the study by Gall K et al. who found that two subsequent xylene washes and incubating it for two hours produces a suitable best quality and quantity of DNA for successful PCR analysis [25]. In the present study, deparaffinization by heating failed to give a better yield of DNA as compared to xylene. Thus far, deparaffinization through heating has used the microwave, boiling, or the thermocycler method; however, we have incorporated a new method of deparaffinization through heating on the hot plate but it was not found to be an efficient method, which is in contrast to studies performed by Banerjee SK et al., Morgan K et al., and Coombs NJ et al. who have used the microwave and thermocycler and found it to be a better method than xylene for obtaining optimal integrity of DNA [12,26,27]. The poor quantity and quality obtained by the heating method in the present study was mainly because prolonged overheating may degrade DNA.

The present study compared two commonly used DNA extraction methods: the conventional phenol–chloroform extraction with HiPurATM paraffin-embedded tissue DNA kit for the archival FFPET group and the long-term FFT group. Results from the present study shows that irrespective of group, DNA isolated through conventional phenol–chloroform gives better mean quantity and quality of DNA in comparison with the DNA kit method. This finding is in favor with studies conducted by Cao W et al. and Liboria TN et al. who compared conventional and kit methods for DNA isolation and showed better quantity and quality of DNA recovered using the conventional phenol–chloroform method. In contrast to the present findings, a study performed by Coombs NJ et al. showed that high-quality DNA was obtained using the kit method instead of the conventional technique [12,28], whereas Chang PKS et al. and Mirmomeni et al. reported that both extraction techniques give suitable DNA templates [2,13].

The phenol–chloroform extraction technique is the most reliable and commonly used protocol for any molecular analysis. This technique also gives freedom to alter concentration and various other factors depending on the samples to be treated. Therefore, it is the more preferred and accepted method for DNA isolation from archival tissue samples which demand a meticulous and effective DNA isolation method. Liboria TN et al. [29] performed an experiment to evaluate the efficiency of the phenol–chloroform method to extract suitable DNA from archival oral tissue samples and concluded that DNA extraction from this method is easy and beneficial in extracting genomic DNA recovered from archived PETs. 

Protein digestion by proteinase K enzyme is considered to be one of the crucial steps in DNA isolation, as proper digestion of the cell surface protein will lead to disruption of the cell membrane which in turn causes the release of cytoplasmic content and breakdown of the nucleus containing target genomic DNA. In the case of formalin-treated samples, protease digestion holds importance as it helps in the breaking of cross-linkages formed by formalin and it also removes formalin salts residue which otherwise may lead to degradation of DNA. In the present study, the elongated digestion procedure was applied by keeping samples dissolved in digestion buffer and proteinase K overnight in incubation at 55–56 °C to ensure sufficient digestion of samples, which was in favor with studies undertaken by Jackson DP et al. [30] and An FS et al. [31], who investigated problems associated with the use of PCR reaction to amplify specific DNA fragments from FFPET and can be possibly due to the presence of inhibitors which will interfere with the functions of the reaction. The results from their experiment indicate that removing the inhibitors can be performed by proteinase K digestion, following which chloroform or phenol purification is performed. In our study, after the precipitation of DNA, further purification was also conducted to remove any residual salts or proteins that may contaminate obtained DNA. Therefore, the extended digestion and purification method used in the present study for the conventional phenol–chloroform protocol will substantiate a better yield of DNA obtained by this method in comparison with DNA isolated through DNA. Thus, the present study findings suggest that the conventional phenol–chloroform technique was the most suitable and effective method to retrieve efficient DNA integrity from archival FFPET or FFT samples.

The present study compared the mean quantity and quality of obtained DNA from Group I and Group II to assess the effectiveness of storing tissue samples and to conclude whether FFPET or FFT is a better medium to obtain sufficient integrity of DNA. Mean DNA quantity and quality of DNA come out to be higher in Group I when in comparison with Group II (Table 3). The findings of the present study were in favor with the study conducted by Romero RL et al. [32] and Niland E et al. [33], who compared integrity from samples stored in formalin with that of FFPET. The study result showed FFPE tissue which was not fixed in formalin for a period of more than three days is a beneficial source for DNA, whereas FF tissue failed to prove a dependable DNA source to perform PCR. 

FFPET comes out to be a better source to obtain DNA in comparison with long-term FFT. This is mainly because long-term storage of samples in formalin will eventually degrade the nucleic acids, whereas the overnight fixation in formalin and embedding in paraffin wax will prevent additional cross-linking of proteins in DNA. Thus, the findings of this study suggest that to obtain suitable and effective quality and quantity of DNA that can be taken further for any molecular analysis it is mandatory to archive tissue samples in overnight formalin fixation and embedded in paraffin wax. 

In our study, all samples in different study groups subsequently being treated through different protocols were subjected to further PCR analysis to evaluate their efficiency through their ability to amplify. Comparing the mean PCR amplification between Group I and II revealed that mean amplification was higher in the FFPET group (Table 9). This is following Romero RL et al., who assessed the usefulness of DNA obtained from FFPET and FFT samples and concluded that FF tissue is not a reliable and effective source of DNA for the PCR technique. There are various causes that can lead to failure of PCR using DNA isolated from formalin-fixed tissue:
i.The generation of DNA–protein cross-linkages resulting in nucleic acid fragmentation due to formaldehyde solution [2,34].ii.Remanent materials such as formalin does inhibit the amplification reaction.iii.The risk of contamination during the manipulation of samples.

When we compared sub-groups of Group I, results showed that maximum amplification was seen in Sub-Group IA (93.33%) followed by Sub-Group IB (86.66%) and Sub-Group IC (66.66%), which suggests that deparaffinization by xylene followed by the conventional phenol–chloroform method is the most suitable protocol for PCR analysis. In study Group II, maximum amplification was seen in Sub-Group IIB (53.33%), more than in Sub-Group IIA (46.66%). Though the amount of amplification was more in Sub-Group IIB, DNA obtained was of degraded quality, forming multiple amplicons of degraded DNA. Among DNA isolation methods, a much higher amount of DNA amplification was obtained by the conventional phenol–chloroform technique than the kit method.

## 5. Conclusions

In conclusion, to obtain suitable integrity of DNA, tissue samples should be stored by fixation in formalin overnight followed by preparation of paraffin tissue blocks, deparaffinization by xylene, and subjection to the conventional phenol–chloroform DNA isolation protocol, which should be the effective method for preservation and to successfully obtain amplifiable DNA copies for archival collection of pathological tissues and retrospective molecular studies.

## Figures and Tables

**Table 1 diagnostics-12-01219-t001:** Details of sample groups and codes.

Group	Sub-Group	Sample Size (*n*)
Group I (Archival paraffin-embedded tissue blocks—24–48 h formalin fixation)	Sub-Group IA (Deparaffinization by xylene followed by the conventional DNA isolation)	15
	Sub-Group IB (Deparaffinization by heating followed by the conventional DNA isolation)	15
	Sub-Group IC (DNA isolation by HiPurATM Paraffin-Embedded Tissue DNA Purification Spin Kit)	15
Group II (Long-term formalin-fixed tissues preserved in formalin)	Sub-Group IIA (DNA isolation by the conventional method)	15
	Sub-Group IIB (DNA isolation by HiPurATM Paraffin-Embedded Tissue DNA Purification Spin Kit)	15

**Table 2 diagnostics-12-01219-t002:** PCR programmed for the amplification of the p53 gene.

Initial Denaturation	Denaturation	Annealing	Extension	Final Extension	Holding Temperature
94 °C	94 °C	60 °C	72 °C	72 °C	4 °C
1 min	30 s	30 s	1 min	10 min	
1 Cycle	40 Cycles			1 Cycle	
**Name of the Primer**	**Concentration Provided by Company (nmol)**			**Stock Concentration**	**Volume of Water Added (µL)**
p53 (F)	4.2			100 µM/µL	42
p53 (R)	4.6			100 µM/µL	46

**Table 3 diagnostics-12-01219-t003:** DNA quantity and quality obtained across the study groups.

Group	Sub-Group	Mean DNA Quantity (ng/µL)	Mean DNA Quality DNA Purity (1.6–1.8)
Group I	Sub-Group IA (*n* = 15)	129.64	66.67%
	Sub-Group IB (*n* = 15)	50.04	46.67%
	Sub-Group IC (*n* = 15)	36.43	26.67%
Group II	Sub-Group IIA (*n* = 15)	31.94	26.67%
	Sub-Group IIB (*n* = 15)	7.526	13.33%

**Table 4 diagnostics-12-01219-t004:** DNA quantity and quality obtained across the deparaffinizing methods.

Group	Sub-Group	Mean DNA Quantity (ng/µL)	*p*-Value ^%^
Group I	Sub-Group IA (*n* = 15)	129.64	(*p* < 0.05) *
	Sub-Group IB (*n* = 15)	50.04	
Group	SUB-GROUP	Mean DNA Quality DNA purity (1.6–1.8)	*p*-value ^#^
Group I	Sub-Group IA (*n* = 15)	66.67%	(*p* > 0.05)
	Sub-Group IB (*n* = 15)	46.67%	

^%^ Assessment by *t*-test; ^#^ assessment by Z-test; * statistical significance at *p* < 0.05.

**Table 5 diagnostics-12-01219-t005:** Comparison of the mean DNA quantity within Group I (FFPET) and Group II (FFT).

Groups	N	Mean DNA (ng/µL)	SD	*p*-Value Assessment ^%^
DNA quantity assessment within FFPET group				
Sub-Group IA	15	129.648	126.385	Sub-Group IA vs. IB (*p* < 0.05) *
Sub-Group IB	15	50.04	46.267	Sub-Group IB vs. IC (*p* > 0.05)
Sub-Group IC	15	36.43	22.517	Sub-Group IC vs. IA (*p* < 0.01) *
DNA quantity assessment within FFT group				
Groups	N	Mean DNA (ng/µL)	SD	*p*-Value Assessment ^#^
Sub-Group IIA	15	31.94	22.499	*p* < 0.001 *
Sub-Group IIB	15	7.526	6.194	

^%^ Assessment by ANOVA; ^#^ assessment by *t*-test; * statistical significance at *p* < 0.05.

**Table 6 diagnostics-12-01219-t006:** Comparison of the mean quantity of DNA obtained between the methods from FFPET and FFT.

Groups	N	Mean DNA (ng/µL)	SD	*p*-Value Assessment ^%^
Assessment based on the conventional extraction method				
Sub-Group IA	15	129.64	126.38	Sub-Group IA vs. IB (*p* < 0.05) *
Sub-Group IB	15	50.04	46.26	Sub-Group IB vs. IIA (*p* > 0.05)
Sub-Group IIA	15	31.94	22.49	Sub-Group IIA vs. IA (*p* < 0.05) *
Assessment based on the kit method				
Groups	N	Mean DNA (ng/µL)	SD	*p*-Value Assessment ^#^
Sub-Group IC	15	36.42	22.51	*p*< 0.001 *
Sub-Group IIB	15	7.526	6.194	

^%^ Assessment by ANOVA; ^#^ assessment by *t*-test; * statistical significance at *p* < 0.05.

**Table 7 diagnostics-12-01219-t007:** Comparison of the mean DNA quality obtained within the group.

Mean Quality of DNA Obtained within the FFPET Group					
GROUP	Pure DNA (1.6–1.8)	Percentage (%)	Z Value	*p* Value	Result
Sub-Group IA	10	66.67	1.09	0.27	Non-Significant
Sub-Group IB	7	46.67			
Sub-Group IA	10	66.67	2.31	0.02	Significant
Sub-Group IC	4	26.67			
Sub-Group IB	7	46.67	1.22	0.26	Non-Significant
Sub-Group IC	4	26.67			
Mean DNA quality of obtained within the FFT Group					
Sub-Group IIA	4	26.67	0.89	0.37	Non-Significant
Sub-Group IIB	2	13.33			

**Table 8 diagnostics-12-01219-t008:** Comparison of the mean quantity of DNA obtained between the methods from FFPET and FFT.

Group	Pure DNA (1.6–1.8)	Percentage (%)	Z Value	*p* Value	Result
DNA quantity assessment based on the conventional extraction method					
Sub-Group IA	10	66.67	1.09	0.27	Non-Significant
Sub-Group IB	7	46.67			
Sub-Group IA	10	66.67	2.31	0.02	Significant
Sub-Group IIA	4	26.67			
Sub-Group IB	7	46.67	1.22	0.26	Non-Significant
Sub-Group IIA	4	26.67			
DNA quantity assessment based on the kit method					
Sub-Group IC	4	26.67	0.89	0.37	Non-Significant
Sub-Group IIB	2	13.33			

**Table 9 diagnostics-12-01219-t009:** PCR amplification in different study groups.

Group	Sub-Group	Amplification in Percentage (%)
Group I	Sub-Group IA	93.33
	Sub-Group IB	86.66
	Sub-Group IC	66.66
Group II	Sub-Group IIA	46.66
	Sub-Group IIB	53.33

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
