# Peer review of "Evaluation and Comparison of Genomic DNA Extraction Methods and PCR Optimization on Archival Formalin-Fixed and Paraffin-Embedded Tissues of Oral Squamous Cell Carcinoma"

_diagnostics, 2022, doi:10.3390/diagnostics12051219_

Round 1

Reviewer 1 Report

The current study is designed to evaluate the efficiency of different deparaffinized methods and DNA extraction methods on the quality & quantity of DNA from FFPET and to assess the effect on DNA from long term formalin-fixed specimens with that of archival paraffin tissue blocks and to assess the viability of the obtained genomic DNA using PCR technique for p53 gene amplification in OSCC.

This study contributes to obtaining good quality DNA for future retrospective studies from samples of Archives of formalin-fixed tissues (FFT) and formalin-fixed paraffin-embedded tissue (FFPET) of Oral Squamous Cell Carcinoma.

Given the objective of the research, I suggest making a workflow of each method or a comparative table that indicates the common and differential steps of each one. This could be included as supplementary material in the methods section of the manuscript.

Minor observations:

  • Line 172: It is suggested to revise the sentence: “1 l of the eluted DNA was taken in the pipette”
  • Line 187: It is suggested to revise the sentence: “2.5. Gel elec2.5 Gel Electrophoresis and analysis of PCR product”

Author Response

Academic editor comments:

Comment1: Kindly note that graph 9 is cited in the main text(Line 411) while it is
not uploaded either in the system or in the Word file.

Response: We wrongly mentioned as graph 9 in the discussion section. In the revised manuscript the graph 9 is no more required.

Comment2: Plus, Table 3 is not cited in the main text. Please revise them as well.

Response: Suggested changes were made and highlighted in the revised version.

==================================================================

Reviewer 1:

The current study is designed to evaluate the efficiency of different deparaffinized methods and DNA extraction methods on the quality & quantity of DNA from FFPET and to assess the effect on DNA from long term formalin-fixed specimens with that of archival paraffin tissue blocks and to assess the viability of the obtained genomic DNA using PCR technique for p53 gene amplification in OSCC.

This study contributes to obtaining good quality DNA for future retrospective studies from samples of Archives of formalin-fixed tissues (FFT) and formalin-fixed paraffin-embedded tissue (FFPET) of Oral Squamous Cell Carcinoma.

Comment1: Given the objective of the research, I suggest making a workflow of each method or a comparative table that indicates the common and differential steps of each one. This could be included as supplementary material in the methods section of the manuscript.

Response:

Thanks for your suggestion. The   workflow was now added to the revised manuscript.

Minor observations:

Comment A: Line 172: It is suggested to revise the sentence: “1 l of the eluted DNA was taken in the pipette”

Response: The suggested changes were made in the revised manuscript

Comment B: Line 187: It is suggested to revise the sentence: “2.5. Gel elec2.5 Gel Electrophoresis and analysis of PCR product”

Response: The suggested changes were now made in the revised manuscript.

Reviewer 2:

The manuscript describes comparison of methods for DNA isolation from fixed tissue samples. Conventional phenol-chloroform extraction or HiPura ATM paraffin-embedded tissue DNA kit isolation methods were compared. The samples were used directly for DNA isolation procedure or a paraffin removal pre-treatment by xylene were applied. The quantity of the isolated DNA was spectrophotometrically determined, and the quality of the isolation was analyzed by PCR reaction. It is concluded that the phenol-chloroform technique resulted in best quality DNA isolation from preserved tissue. There are some issues as listed below:

 Comment1: Page 5: Please remove “elec2.5”

Response: The suggested changes were now made in the revised manuscript.

Comment2: I think it would be useful to have DNA isolation amounts for freshly prepared tissue as a control for the groups in Table 3 and 4.

Response: We accept with the reviewer. But this present study was designed keeping “Archival” samples as the epicentre, therefore fresh samples as a control group could not be taken into consideration. Adding the fresh sample will divert the objective/focus of the study..

Comment 3: Section “3.1. Mean DNA quantity and quality of obtained DNA from different study groups:” is missing the text.

Response: The suggested changes were now made in the revised manuscript.

Comment 4: Table 5 can be merged into Table 4. Similarly, Table 8, Table 11, Table 14 can be merged into Table 7, Table 10, Table 13 respectively.

Response: The tables were merged as suggested. The tables in the current version were better presented with the required variables for clear understanding of manuscript.

Comment 5: Page 9: “Few studies have shown time when RNA, DNA and nucleic acids extracted from….” Please remove “RNA, DNA” or “nucleic acids”.

Response: The suggested changes were now made in the revised manuscript.

Comment 6: First paragraph of discussion section does not make sense, and it can be removed.

Response: The first paragraph in the discussion section was removed.

Comment 7: Page 11: Table 17& Graph 9 were not provided in the manuscript.

Response: We wrongly mentioned as Table 17& Graph 9 in the discussion section. In the revised manuscript the Table 17& Graph 9 are no more required.

========================================================================

Reviewer 2 Report

The manuscript describes comparison of methods for DNA isolation from fixed tissue samples. Conventional phenol-chloroform extraction or HiPura ATM paraffin-embedded tissue DNA kit isolation methods were compared. The samples were used directly for DNA isolation procedure or a paraffin removal pre-treatment by xylene were applied. The quantity of the isolated DNA was spectrophotometrically determined, and the quality of the isolation was analyzed by PCR reaction. It is concluded that the phenol-chloroform technique resulted in best quality DNA isolation from preserved tissue. There are some issues as listed below:

Page 5: Please remove “elec2.5”

I think it would be useful to have DNA isolation amounts for freshly prepared tissue as a control for the groups in Table 3 and 4.

Section “3.1. Mean DNA quantity and quality of obtained DNA from different study groups:” is missing the text.

Table 5 can be merged into Table 4. Similarly, Table 8, Table 11, Table 14 can be merged into Table 7, Table 10, Table 13 respectively.

Page 9: “Few studies have shown time when RNA, DNA and nucleic acids extracted from….” Please remove “RNA, DNA” or “nucleic acids”.

First paragraph of discussion section does not make sense, and it can be removed.

Page 11: Table 17& Graph 9 were not provided in the manuscript.

Author Response

(The authors gave the same response as above.)

Round 2

Reviewer 2 Report

The revised manuscript is improved according to reviewer suggestions by including some details in the method section and rearranging the contents of the tables.